# Fabrication and Characterization of Biplasmonic Substrates Obtained by Picosecond Laser Pulses

**Andrei Stochioiu [1,2,\*], Catalin Luculescu [3], Irina Alexandra Paun [3,4], Luiza-Izabela Jinga [1] and Constantin Stochioiu [5]**

1. National Institute for Laser, Plasma and Radiation Physics (INFPLR), Atomiștilor 409, RO-077125 Măgurele, Romania; izabela.jinga@inflpr.ro
2. Faculty of Physics, University of Bucharest, Atomiștilor 405, RO-077125 Măgurele, Romania
3. Center for Advanced Laser Technologies (CETAL), National Institute for Laser, Plasma and Radiation Physics (INFPLR), Atomiștilor 409, RO-077125 Măgurele, Romania; catalin.luculescu@inflpr.ro (C.L.); irina.paun@inflpr.ro (I.A.P.)
4. Faculty of Applied Sciences, University Politehnica of Bucharest, Splaiul Independenței 313, RO-060042 Bucharest, Romania
5. Faculty of Industrial Engineering and Robotics, University Politehnica of Bucharest, Splaiul Independenței 313, RO-060042 Bucharest, Romania; c.stochioiu@upb.ro
* Correspondence: andrei.stochioiu@inflpr.ro; Tel.: +40-727-582-654



**Featured Application: Fabricated structures can be used as substrates for the SERS technique of Raman spectroscopy.**

**Abstract:** Bimetallic nanostructures have the potential to become the new generation candidates for applications in catalysis, electronics, optoelectronics, biosensors and also for surface-enhanced Raman Spectroscopy (SERS). The bimetallic nanocrystals offer additional properties over the single metal components such as improved electromagnetic properties and corrosion protection. This work presents a simple and inexpensive method to fabricate large area biplasmonic (bimetallic) substrates, employing DC magnetron sputtering, picosecond laser pulses and a digital galvanometric scanner. The aim of this study was to achieve large area homogeneous substrates while having a good and predictable signal amplification by SERS effect. Gold thin films with 200 nm thickness were deposited on optical polished substrates and then irradiated in atmospheric air with λ = 1064 nm wavelength laser pulses with 8 ps pulse duration and 500 kHz fixed repetition rate. Various laser fluences and laser irradiation speeds were employed in order to optimize the Laser-Induced Periodic Surface Structures (LIPSS) formed on the substrate. The results are presented comparatively for the standalone Cu substrates and for the Cu-Au substrates using Raman spectral analysis on a single signal peak of a Rhodamine 6G solution.

**Keywords:** SERS; laser ablation; Raman microspectroscopy; bimetallic structures; LIPSS

---

## 1. Introduction

Surface nanostructuring using laser irradiation has always proven to be an impactful field of study. The electromagnetic properties of nanostructures and the way they interact with incident photons can provide improvements to the properties of bulk unprocessed materials [1]. By employing a high power picosecond laser to process bimetallic structures instead of a femtosecond laser, cheaper products can be obtained. In addition, the pulse duration of a few picoseconds is long enough to induce some amount of thermal ablation, which might force the two metals to diffuse [2].

Surface-Enhanced Raman Spectroscopy, also known as Surface-Enhanced Raman Scattering (both SERS), is one of the many Raman spectroscopy variants. SERS is a phenomenon in which the Raman scattering of the specific molecular vibrations is enhanced in the proximity of the nanostructured metal substrate on which the analyte is placed [3]. Many types of nanometric structures were employed successfully in the field of SERS: nanoparticles [4–7], nanorods [8], nanocones [9], nanoholes and nanodisks [10]. Moreover, many methods of obtaining SERS-active nanostructures have been tested and proven throughout the years. Just to name a few: self-assembling techniques [11], electron beam lithography, ion beam irradiation [12], nanotransfer printing [13], femtosecond laser irradiation and picosecond laser irradiation [14–17].

This paper aims to improve the fabrication process of SERS-active substrates with periodic nanostructured surface using ultrashort high-intensity laser pulses. The novelty of this study involves mixing two already proven SERS-active metals, in this case in the form of a thin film of gold on top of copper, followed by laser irradiation, providing cheap and easily scalable production steps.

Laser-Induced Periodic Surface Structures (LIPSS) are sub-wavelength structures obtained by laser irradiation [18]. They can be observed on the surface of many materials after femtosecond or picosecond laser irradiation and, for the purpose of this work, they are used as substrates for enhancing the Raman signal intensity used in Raman Spectroscopy [14–16]. In addition, it was demonstrated, both analytically and numerically, that bimetallic structures could increase the electromagnetic field more than single-metal counterparts [4,19], which suggests that employing a bimetallic structure might yield better results than each of them alone. Moreover, at some values of thin film thicknesses and laser fluence, the laser ablation should not only drill holes into the thin film, but should also melt it and allow it to diffuse into the layer below [20]. In this work, bimetallic submicron-sized structures, which will act as reliable SERS substrates, were built by employing laser ablation as a tool. These above-mentioned bimetallic structures would expand the availability of SERS substrates.

## 2. Materials and Methods

### 2.1. Materials

Gold (PI-KEM Ltd., Tamworth, UK, 99.99%) and copper (Color Metal Ltd., Mogoșoaia, Romania, standard C11000 [21]) were used as materials for substrate fabrication. Rhodamine 6G (Voigtländer Kriminaltechnik GmbH, Riedböhringen, Germany, 99.9% powder, abbreviated as R6G) was used as analyte for the Raman characterization of the substrates, either as out-of-the-box powder, or dissolved in distilled water.

### 2.2. Methods

Plate preparation:
High purity copper was used for the study in the form of 3 mm thick disks of 60 mm diameter both as standalone material or in combination with gold.

High purity gold was used in magnetron sputtering equipment (Cresington 108) as target for the bimetallic study. In this way, a thin film of gold was deposited on the polished copper surface.

The highly fluorescent dye R6G was used as pure powder for the basic Raman spectroscopy. An aqueous solution of R6G with $10^{-4}$ M molar concentration was prepared using distilled water at room temperature for the SERS amplification study.

The Cu plates were optically polished and then washed with ethanol after a 10 min sonication to remove impurities from the surface. After this, one plate was coated with a thin film of 200 nm thickness of Au with the DC magnetron sputtering system. Further mentions of this resulting product of bulk Cu coated with 200 nm of Au are called simply Cu-Au. Thin film thickness measurements were taken during the film deposition using a thickness measurement system equipped with a quartz crystal, as provided by the deposition system. Both coated and uncoated plates were then laser irradiated.

A Lumera Hyperrapid 50 (diode pumped Nd:YAG laser), with 1064 nm wavelength, was used for the substrate fabrication. The laser system delivered pulses of 8 ps duration, fixed 500 kHz pulse repetition rate and adjustable average power up to a maximum of 50 W, corresponding to 100 μJ average pulse energy, with a Gaussian (TEM00) energy distribution and beam quality factor ($M^2$) higher than 1.4, according to the datasheet. The laser beam was directed using a galvanometric scanner and focused on a circular spot of 72.5 ± 3.5 μm fixed diameter (value at $1/e^2$ of the maximum intensity of the Gaussian profile of the laser beam [20]) by an f-theta lens where f = 160 mm for all experiments.

Laser ablation experiments:

Laser markings on the plate were created by ablating tightly packed parallel lines, which formed an ablated square of 3 by 3 mm. A total of 49 such squares were laser ablated onto the surface of each plate, arranged into a 7 by 7 matrix. Each row of squares was ablated using a different laser fluence and each column, with a different laser marking speed. This way, each plate was ablated with 49 squares formed with different combinations of laser fluence and laser writing speeds.

The laser fluences range from (0.3 ± 0.02) J/cm$^2$ up to (2.42 ± 0.22) J/cm$^2$. Variation of the laser fluence was achieved by using the power setting provided by the control system, which varied the pulse energy in steps of 0.3 J/cm$^2$. The value of 1.8 J/cm$^2$ was skipped.

The laser marking speeds ranged from 100 mm/s to 1000 mm/s. Considering that the pulse repetition rate is a constant 500 kHz, the laser spot diameter is fixed approximately at 72.5 μm and the trajectory is always linear, the laser writing speed parameter is described in its equivalent average number of overlapping pulses. It expresses the average number of pulses of the calculated spot diameter which would overlap on any given point within the linear trajectory of the laser markings. A higher laser writing speed will yield a lower number of overlapping pulses and vice versa. The lowest laser writing speed, 100 mm/s, is equivalent to approximately 360 overlapping pulses, while the highest writing speed, 1000 mm/s, is equivalent to approximately 36 overlapping pulses. Intermediary laser writing speeds were in increments of 150 mm/s, producing a total of 7 different laser writing speeds. Consequently, the calculated values for the number of overlapping pulses were approximated as follows: 36, 42, 51, 65, 90, 144 and 360.

Morphology and composition characterization:

The resulting nanostructures, then, had their macro and nano topology characterized through Scanning electron microscopy (SEM) and their chemical composition characterized through Energy-dispersive X-ray spectroscopy (EDX) and X-ray photoelectron spectroscopy (XPS).

The substrates were first observed under a scanning electron microscope (FEI Inspect S50) with no tilting at an operating voltage of 25 kV and photographed at a magnification of 20,000. EDX single spectrum was obtained at an accelerating voltage of 8 kV and 5000 magnification and the mapping was obtained at 2.5 kV and 2500 magnification.

Due to the fact that the XPS instrument had a hemispherical analyzer, the electrons, which are allowed to pass and arrive to the detector slits, had a given energy called pass energy.

Spectral resolution depends on the values of: pass energy (lower values give higher resolution), number of scans (increased number of scans increases the peak signal) and the duration of each scan (dwell time). For the XPS survey spectra, the parameters used were: pass energy 50 eV, 10 scans, dwell time 50 s, while for high-resolution spectra: pass energy 10 eV, 50 scans and dwell time 200 s.

SERS characterization experiments:

The SERS effect on the Raman signal of the R6G on the LIPSS generated was assessed by single point Raman microspectroscopy. The fluorescence of R6G is considered noise during Raman Spectroscopy and attempting to amplify the Raman signal through an increase in exposure time will also amplify the fluorescence intensity.

In order to test the Raman signal enhancement, a diluted R6G solution was pipetted onto every substrate, in drops of 25 μL. Considering that the volume of R6G excited by the laser spot is the same for every substrate, it is safe to assume that the same number of R6G molecules are excited by the Raman spectroscopy laser. The drops were then left to dry onto the surface.

All Raman spectra of R6G solution with $10^{-4}$ M molar concentration were collected using 10 s of laser exposure time per acquisition and by calculating an average over 10 spectral acquisitions, with a total of 100 s of exposure time and a spectral resolution of 3.4 cm$^{-1}$. A 632.8 nm HeNe laser with 2.2 mW average power was used for all Raman spectroscopy experiments and was focused onto the substrates using a 100× objective, which reduced the laser spot diameter to 2 μm.

## 3. Results

For the fabrication process, the simple procedure was enough to create a surface nanostructure, which was then capable of Raman signal enhancement of the R6G spectrum via SERS effect.

Through SEM, the formation of repetitive nanostructured ripples on most of the fabricated substrates was observed. An overview of the laser fluence and the number of overlapping pulses on LIPSS created on the Cu substrate covered by 200 nm Au thin film is presented in Figure 1. The effect of the laser fluence on LIPSS formation is visible on the horizontal axis. It can be observed that the ripples become more pronounced with increasing laser fluence. In addition, the effect of the number of overlapping laser pulses on LIPSS formation is presented in the same figure. At much higher laser fluence and larger numbers of overlapping pulses, the periodicity will vanish, as a large amount of material is ablated. A "sweet spot" for ripples formation can be found by considering the two variables.

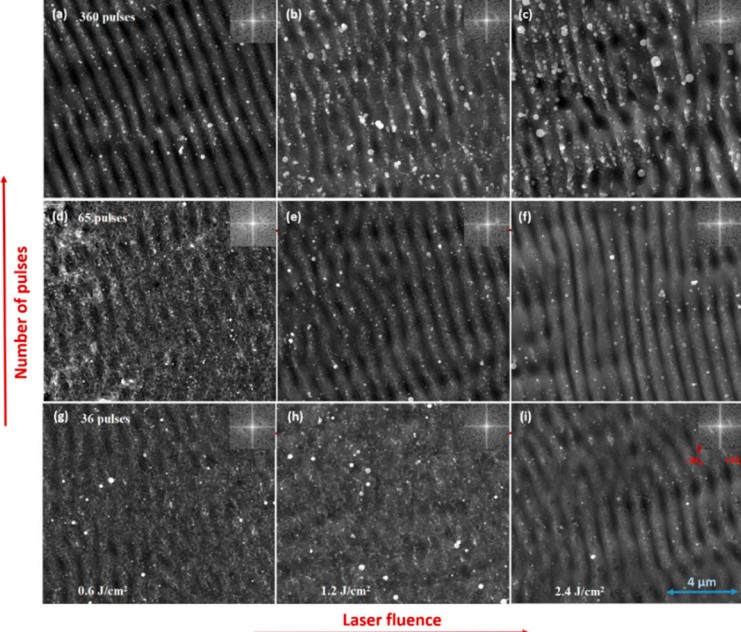

**Figure 1.** SEM micrographs (20,000×) showing the influence of laser fluence and number of overlapping laser pulses on Laser-Induced Periodic Surface Structures (LIPSS) formed on Cu-Au. The insets show the intensity of the Two-Dimensional Fast Fourier Transform (2D FFT) transform and their scale is $k_0 = 5.9052 \, \mu m^{-1}$. (**a**) 0.6 J/cm$^2$, 360 pulses; (**b**) 1.2 J/cm$^2$, 360 pulses; (**c**) 2.4 J/cm$^2$, 360 pulses; (**d**) 0.6 J/cm$^2$, 65 pulses; (**e**) 1.2 J/cm$^2$, 65 pulses; (**f**) 2.4 J/cm$^2$, 65 pulses; (**g**) 0.6 J/cm$^2$, 36 pulses; (**h**) 1.2 J/cm$^2$, 36 pulses; (**i**) 2.4 J/cm$^2$, 36 pulses.

In order to obtain the surface-pattern period, a Two-Dimensional Fast Fourier Transform (2D FFT) from each SEM micrograph was performed. The 2D FFT image is shown on the right-hand-side of each SEM micrograph of Figure 1. Only the magnitude/intensity part of 2D FFT was used. The analysis performed on the high magnification SEM pictures revealed the period of the nanometric ripples, which was around 800 nm. The scale for the 2D FFT insets from Figure 1 is given in wave numbers, calculated with $k_0 = 2\pi/\lambda_0$, where $\lambda_0 = 1064$ μm is the wavelength of the laser pulses.

From the experiments performed under various laser fluences and number of overlapping pulses it seems that the morphology of LIPSS is largely influenced by the laser energy density delivered on the unit of area. It is visible that the laser fluence and number of overlapping pulses should exceed the values used to produce (d) and (h) in order to obtain LIPSS. As shown in Figure 1d,g,h, both the laser fluence and number of overlapping pulses have their own threshold, which has to be passed in order to obtain defined LIPSS. At the higher values for both variables, the periodicity is affected, most probably by a different regime of laser ablation. The 2D FFT images reveal that these values must not pass the ones used to produce (b) and (f).

The SEM image of one laser processed Cu-Au portion is presented below, in Figure 2a. A wide area of regular ripples was formed in the laser irradiated region, visible as a darker vertical stripe. EDX surface mapping analysis indicated the presence of small quantities of Au on the ripples area (Figure 2e) compared to the quantity of Cu. The more in-depth probing of the EDX technique goes well beyond the 200 nm thin film of Au, therefore even a small percentage is meaningful after laser processing. Additionally, XPS survey spectra shown in Figure 2b,d also revealed the presence of Au on the laser processed surface but, as expected, in considerably smaller amounts than the unprocessed surface. Due to the laser ablation, small amounts of Cu are also present on the unprocessed surface.

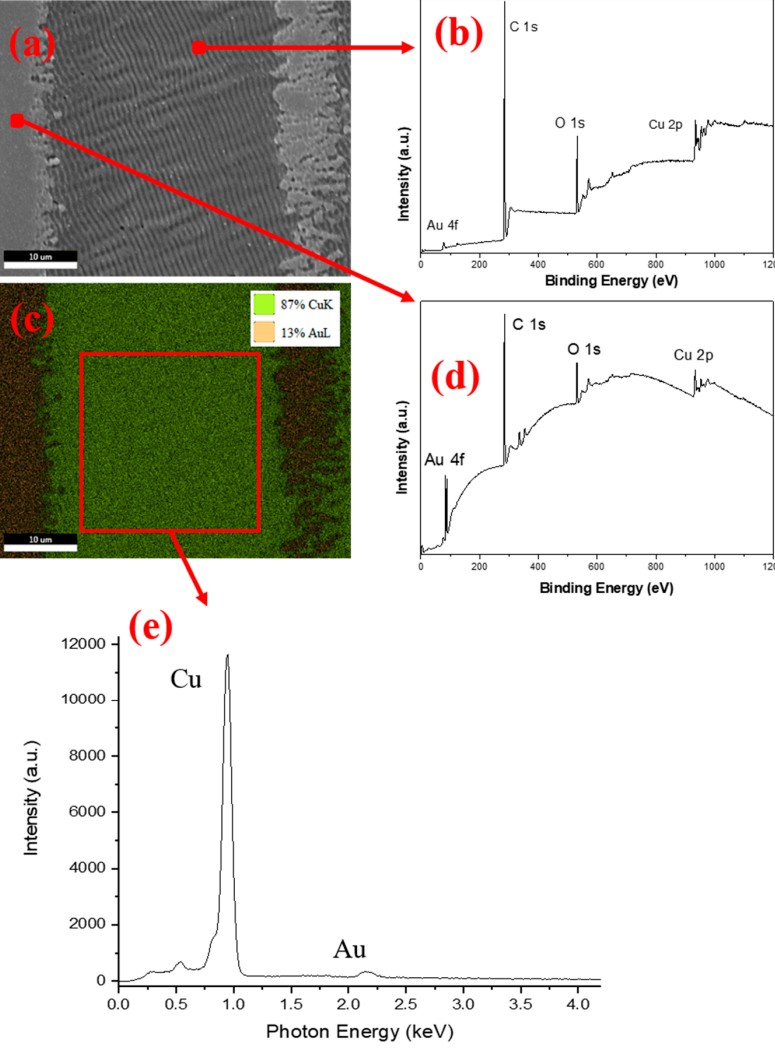

**Figure 2.** SEM image (**a**) and EDX mapping (**c**) of a Cu-Au substrate produced by laser irradiation with 0.6 J/cm$^2$ laser fluence, 360 overlapping pulses; XPS Surveys spectra of processed (**b**) and unprocessed points (**d**); EDX spectrum of processed area (**e**).

Furthermore, the XPS experiments revealed how the quantity of Au in the composition changes by varying the number of overlapping pulses or the laser fluence. In Figure 3, the intensity of the Au peak at 84 eV falls rapidly, both by increasing the number of overlapping pulses and by increasing the laser fluence. The point of zero laser fluence in the right hand panel represents the XPS intensity of an unprocessed area of Cu-Au.

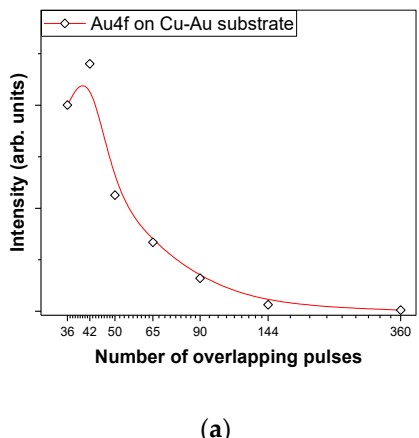 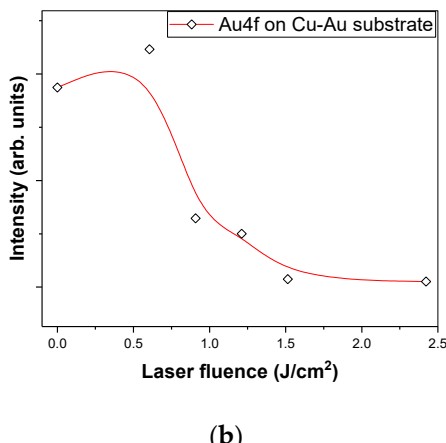

(**a**)　　　　　　　　　　　　　　　　　　　　　　　(**b**)

**Figure 3.** XPS Intensity variation of the 84 eV peak of Au on substrates produced by picosecond laser irradiation with either different laser fluences and fixed 65 overlapping pulses (**a**) or produced with different number of overlapping pulses and fixed 1.2 J/cm$^2$ laser fluence (**b**); red lines are B-spline interpolations.

In Figure 4, an experimental characterization of the high purity R6G powder's Raman spectrum is presented. The high intensity peaks from literature can be observed [21–24]. After having applied the built-in fluorescence correction software [25], the peaks are close to the background noise. It is also difficult to draw sufficient information from the Raman fingerprint regions. It is worth stating that, for R6G, increasing the laser power or the spectral data acquisition time causes both the Raman signal and the fluorescence signal to intensify without significant changes to their ratio. A time consuming method of increasing the signal-to-noise ratio is to irradiate the sample without gathering data and wait for the fluorescent response to lose some of the initial intensity.

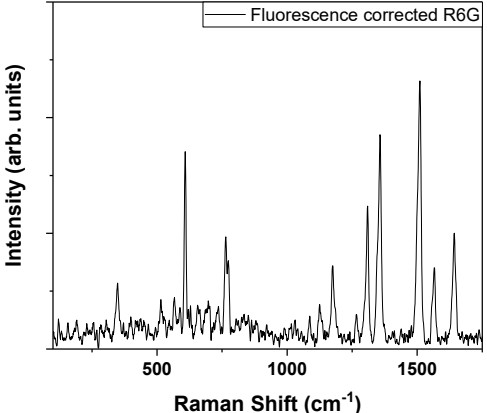

**Figure 4.** Raman spectra of pure Rhodamine 6G solution (R6G) powder on glass after fluorescence correction.

Further on, Raman signal amplification test results are presented below as direct comparison between different samples of Cu and Cu-Au substrates. In Figure 5, the corrected Raman spectra of the first analyzed substrates are presented. These are Cu and Cu-Au, which were produced with the

same laser fluence of 1.2 J/cm$^2$, but with different average numbers of overlapping pulses. On the left hand panel, the Cu-Au substrates appear to provide satisfying SERS amplification, considering that the amount of substance excited by the Raman laser is considerably smaller because of the lower concentration of the analyte ($10^{-4}$ M solution compared to 99.9% purity powder). For better visibility, the intensity of a single peak for each substrate was also plotted in Figure 5. It shows the Raman intensity of the most intense Raman shift value for R6G excited with 632.8 nm laser (caused by C-C and C-N stretching [21–24]), i.e., the 1356 cm$^{-1}$ peak of each spectrum. The detailed analysis of the maximum peak of R6G from 1356 cm$^{-1}$ reveals that between 50 and 90 overlapping laser pulses is the optimum from the point of view of SERS effect. Judging by the visibly distorted ripples seen in Figure 1d,g,h, the slightly less powerful amplification at low numbers of overlapping pulses could be caused by the lack of well-formed ripples. For Cu, the higher number of overlapping pulses seems to provide worse enhancement. The laser fluence of 1.2 J/cm$^2$ might have been too high over the ablation threshold and so the variation of amplification is not significant when varying the number of overlapping pulses which produced the substrate. The difference between the Cu and the Cu-Au substrates peak intensities (i.e., a difference higher than an order of magnitude) generated by the variation of the number of overlapping pulses is not visible. In addition, higher laser fluences and numbers of overlapping pulses lead to the formation of oxide droplets, which do not increase the SERS amplification.

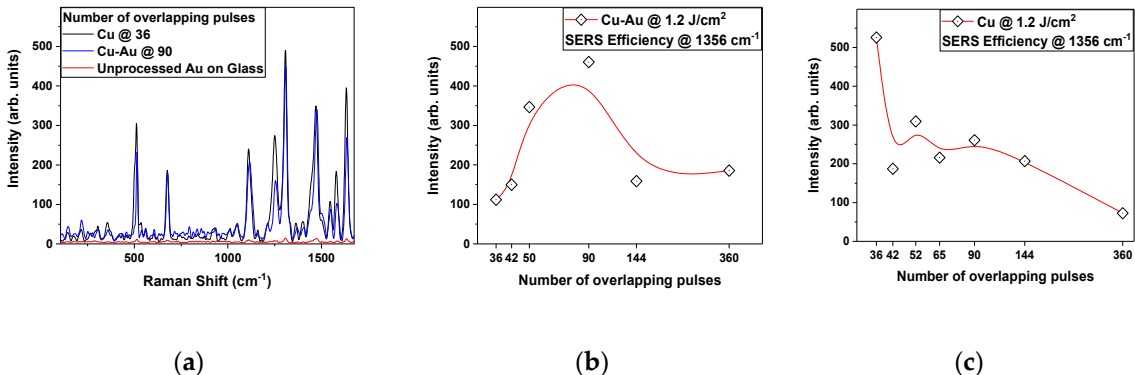

(**a**)       (**b**)       (**c**)

**Figure 5.** Raman spectra for R6G with $10^{-4}$ M concentration on Cu-Au, Cu and unprocessed Au substrates (**a**); surface-enhanced Raman Spectroscopy (SERS) efficiency for Cu-Au substrates (**b**) and Cu substrates (**c**) on the 1356 cm$^{-1}$ peak of R6G with $10^{-4}$ M concentration, fabricated with different numbers of overlapping pulses and the same laser fluence of 1.2 J/cm$^2$; red lines (middle and right panels) are B-spline interpolations.

Another analysis was made on the same plates by comparing the results obtained from the substrates produced by varying the laser fluence, while maintaining a constant number of 50 overlapping pulses. In Figure 6, the spectra obtained for Cu-Au (b) and Cu (c) are presented. Apart from the strong signal of R6G on the Cu-Au substrate processed with 0.6 J/cm$^2$ fluence, for all the other fluences investigated there is no clear dependence of the SERS intensity on the laser fluence.

The highest peak intensity was obtained for Cu at a fluence of 1.2 J/cm$^2$. For Cu-Au, a highly distinguishable intensity peak was obtained at 0.6 J/cm$^2$, as all the other fluences produced visibly lower amplification. There is a connection between the XPS intensity of Au (Figure 3) and the Raman signal intensity when varying the laser fluence (Figure 5), but the connection is nonexistent when varying the number of overlapping pulses in the case of Cu-Au substrates (Figure 6).



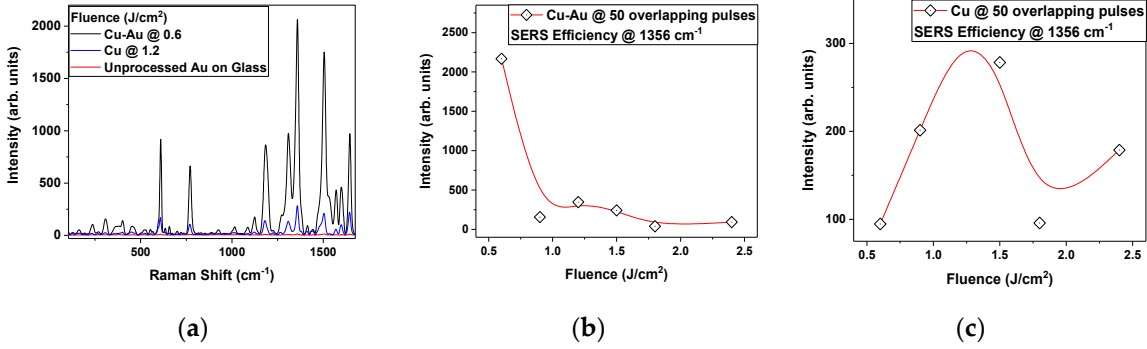

**Figure 6.** Raman spectra for R6G with $10^{-4}$ M concentration on Cu-Au substrates (**a**); SERS efficiency for Cu-Au substrates (**b**)and Cu substrates (**c**) on the 1356 cm$^{-1}$ peak of R6G with $10^{-4}$ M concentration, fabricated with different laser fluences and fixed 50 overlapping pulses; red lines (middle and right panels) are B-spline interpolations.

Following this analysis, the best substrate was selected and its performance was compared with a commercial product for validation of the amplification results. The comparison product was chosen from SILMECO Gold SERS series [26,27], since the producers also use Au nanostructures for SERS.

In this brief comparison, R6G spectral analysis was conducted and the result is presented in Figure 7—the highest intensity spectrum represents the commercial product SILMECO Au, the red line represents the best substrate from the present work (Cu-Au produced with 0.6 J/cm$^2$ laser fluence and 36 overlapping pulses) and the blue line represents the best Cu substrate (1.2 J/cm$^2$ and 36 overlapping pulses). The almost flat pink line represents an unprocessed Au substrate (thin film of Au on glass). Even though the same drop that contained R6G was also pipetted onto a thin film of Au, the lack of substrate processing led to insignificant amplification.

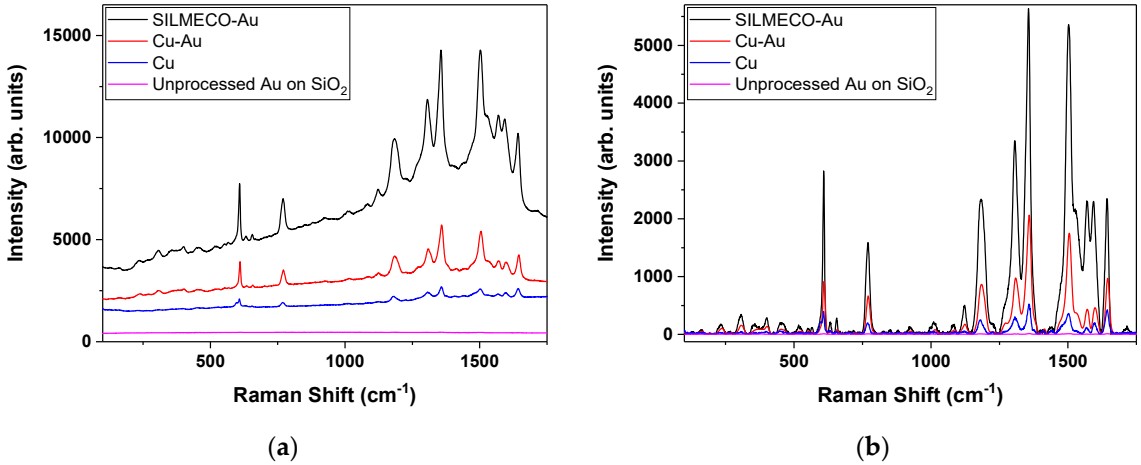

**Figure 7.** Comparison of Raman spectra for R6G with $10^{-4}$ M concentration on SILMECO-Au, Cu-Au (200 nm) laser ablated, Cu laser ablated and a flat Au substrates acquired in the same conditions: (**a**) raw spectra; (**b**) fluorescence corrected spectra.

By combining Cu and Au in a biplasmonic substrate, a superior signal amplification is observed, compared with the simple Cu substrate. The Raman signal is sufficiently amplified by the SERS effect given by the use of Cu-Au substrate, providing that the Raman signal intensities are just a few times lower than a commercial substrate.

## 4. Conclusions

The present work investigates the laser processing of materials in order to obtain bimetallic SERS substrates. A picosecond laser was used to produce bimetallic submicronic structures on copper plates, either as they were or coated with a thin film of Au. This procedure resulted in diffusion mixing of two metals during the ultrashort laser ablation and low oxidation, which could be an explanation for the promising results obtained.

Several substrates were fabricated by using the proposed technique and their quality was evaluated through direct Raman spectroscopy of Rhodamine 6G. It has been shown that the biplasmonic Cu-Au substrate has the best performance when compared with the simple Cu substrate, but it is inferior when compared to the commercial SILMECO Au.

**Author Contributions:** Conceptualization: A.S., C.L., I.A.P.; methodology: A.S., C.L.; software: A.S.; investigation, data curation, formal analysis and validation: A.S., C.L. and L.-I.J.; writing—original draft preparation: A.S., C.L.; writing—review and editing: A.S., C.L., I.A.P., C.S. and L.-I.J.; visualization: L.-I.J. and C.S.; resources, funding acquisition, project administration and supervision: C.L. and I.A.P. All authors have read and agreed to the published version of the manuscript.

**Funding:** This work was supported by Romanian Ministry of Education and Research, under Romanian National Nucleu Program LAPLAS VI—contract n. 16N/2019 and the projects 197PED/2017 and 218PED/2017 from UEFISCDI and the doctoral scholarship from University of Bucharest, Faculty of Physics.

**Acknowledgments:** Special thanks are given to Răzvan Ungureanu, INFLPR, CETAL Department.

**Conflicts of Interest:** The authors declare no conflict of interest.

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
