# Peer review of "Fabrication and Characterization of Biplasmonic Substrates Obtained by Picosecond Laser Pulses"

_applsci, doi:10.3390/app10175938_

Round 1

Reviewer 1 Report

This paper describes the fabrication of LIPSS formed bimetallic Cu-Au substrates and its application to SERS. This paper has interests in bimetallic nanostructure. However, the physical consideration and the advantage of the use of the bimetallic structure is unclear. Unfortunately it must be concluded that lack of scientific novelty and advancement of knowledge in this manuscript make it unsuitable for publication.

  • They don't refer significant articles related on SERS substrates formed by LIPSS.[*] Ann.Phys. 524, 11, L5-L10 (2012) [**] ACS Sens. 1, 318-323 (2016)
  • Commercially available SERS substrates SILMECO-Au exhibits best enhancement of SERS. If the author want to insist the performance of Cu-Au bimetallic SERS substrate, they shoud show the SERS result not unprocessed Au on SiO2 but laser processed Au on SiO2.
  • In Fig.5, 50 and 90 overlapping laser pulses show the high intensity of SERS signal, but the author has no physical consideration of thie effect. It is better to consider about this reason or to show average plots with standard deviation for the reliability of this result.
  • This manuscript unsatisfies the scientific paper style. In page 5, line 174 (Fig.2); it is lack of the result of weight distribution of Cu (mentioned in the last part of caption). In Fig.4, No result shows the left panel of Raw Raman spectra of pure R6G powder on glass. Typographic errors Page 5, line 175: "atfer" -> "after", Page 8, line 234: "mantaining" -> "maintaining"

Reviewer 2 Report

The authors of the manuscript under review report the fabrication and characterization of biplasmonic substrates by using picosecond laser pulses. They analyze and optimize the operating parameters of the picosecond laser source for biplasmonic substrates fabrication. I think the obtained results are novel, and the technical discussion of this manuscript is of high quality. The only suggestion is to add a theoretical discussion to support the experimental observation. Overall, I would recommend the manuscript to be published in Applied Sciences. 

Reviewer 3 Report

This manuscript reports a LIPSS processed bimetallic substrate for Raman SERS amplification propose purpose which is interesting and rarely reported elsewhere. The method is simple and cost effective. I would recommend for publish after addressing the following issues.

  1. Figure 1, I would suggest to add the actual number of pulses and laser fluence at the left side and bottom of the figure 1

  1. Figure 2, Scale bar is missing for figure 2c.

  1. Figure 4, left panel missing, also here, instead of using glass substrate, I would suggest to use MgF2 CaF2 or quartz since glass produce very high Raman background, you may barely see the R6G signal.

  1. Line 146, Endnote error.

  1. Why authors chose 36, 65 and 360 overlapping pluses rather than other numbers?

  1. To prove the betterness of the method, authors should also compare the enhancement factor of their method with other published results refer to LIPSS for SERS, for example how much better of using bimetalic method compared with mono-metalic method.

Round 2

Reviewer 1 Report

The authors well explained and appropriately modified the manuscript according to each reviewers comments. I would like to recommend this revised manuscript to be published in Applied Sciences as it is.